# Impact of COVID-19 Pandemic on Utilization of Healthcare Services and Spending Patterns in Dubai, United Arab Emirates: A Cross-Sectional Study

**DOI:** 10.3390/healthcare12040473

**Published:** 2024-02-14

**Authors:** Meenu Mahak Soni, Heba Mohammed Mamdouh, Eldaw Abdalla Suliman

**Affiliations:** 1Health Economics and Insurance Policies Department, Dubai Health Authority, Dubai P.O. Box 4545, United Arab Emirates; 2Department of Data Analysis, Research and Studies, Dubai Health Authority, Dubai P.O. Box 4545, United Arab Emirates; hmmohammed@dha.gov.ae; 3Department of Strategy and Governance, Dubai Health Authority, Dubai P.O. Box 4545, United Arab Emirates; easuliman@dha.gov.ae

**Keywords:** COVID-19, Dubai, healthcare, spending, services, utilization, United Arab Emirates

## Abstract

Background: The COVID-19 pandemic affected the utilization of healthcare services in many parts of the world. The response to the healthcare burden imposed by the COVID-19 pandemic was associated with ensuring the provision of optimum healthcare services. This study aimed to estimate the effect of the COVID-19 pandemic on health services utilization and spending patterns in Dubai, the UAE. Methods: This cross-sectional study used secondary data on healthcare utilization and spending to compare between 2019 and 2020. The data was extracted from the health insurance claims on the eClaimLink platform. Descriptive and inferential statistics were used to calculate the percent change of service utilization and spending and percentages of total claims by each encounter type across major diagnostic categories (MDCs). Results: In 2020, there was an overall reduction in outpatient visits and inpatient admissions of 27% and 21%, respectively, compared to 2019. Outpatient visits and prescriptions decreased across all the MDCs except mental and behavioral disorders, which showed an increase of 8% in outpatient visits and 29% in prescriptions. The admissions to the healthcare facilities were also reduced significantly across various MDCs, ranging from 10% to 44%. Similarly, a downward trend was seen in diagnostics for different MDCs. An increase in expenditure on diagnostics and drugs for neoplasm was reported, despite a corresponding decrease in outpatient and inpatient admissions for the same. Conclusion: A significant decrease in overall healthcare utilization and corresponding healthcare spending, resulting from a decline in outpatient and inpatient volume in healthcare facilities at all the levels (hospitals, clinics, speciality centres), was reported during the pandemic. The impact of the pandemic on mental health was reported by this study, as it showed an upward trend in utilization and spending. For the neoplasms, although the utilization declined, the expenditure on diagnostics and drugs during each encounter increased significantly.

## 1. Introduction

The coronavirus disease (COVID-19) has not only exacted a tragically high toll on human lives, but it has also come at an unprecedented, enormous financial cost [1]. Several governments have spent large sums of money on tackling COVID-19, and societal problems resulted from their reactions to it. Research has indicated a total of $10 trillion (U.S.) was dispersed as government stimulus packages in the months following the start of the pandemic government [2]. There were also countries who spent far less [3].

Worldwide, COVID-19 caused an unprecedented shift in healthcare utilization. Research has shown that people missed medical care due to various reasons, including inability to access services as a result of lockdown policies, fear of getting an infection while visiting a care facility, and suspension and cancellation of services such as elective surgeries [4]. In Europe, for example, policy responses to COVID-19 led some countries to act swiftly to mobilize extra funds and minimize financial barriers to healthcare access. The system advocated for higher government spending on healthcare [5]. In addition, due to the pandemic, changes in healthcare systems increased clinicians’ risk of medical litigation in some settings, with higher risks related to the decision-making in cased of patients with confirmed or suspected COVID-19 [6]. In such situations, malpractice could have impacted both healthcare spending and quality [7].

International data also showed that the continuity of some essential health services in many countries was affected by the government’s policies and measures that were implemented during the years of the pandemic. Service disruptions included inpatient or outpatient services and community-based care in almost half of the countries surveyed [8]. Globally, service disruption ranged from 70% to 55% of the normal performance. The most significant disrupted services included cancer diagnosis and treatment (55%), noncommunicable diseases, including cardiovascular diseases, by 69%, routine immunization services (70%), and mental health disorders (61%) [8]. Data indicated the disruption in healthcare services had observed adverse health outcomes at many levels [9]. On the other hand, the pandemic may also have resulted in some people being spared unnecessary care that has the potential to cause harm [9,10]. Notwithstanding, the impact of the pandemic has varied markedly according to the location and type of the facility and the type of the healthcare system [11,12,13,14,15]. In Romania, for example, a nationwide analysis revealed that the COVID-19 epidemic had a negative financial impact and significantly reduced the number of arthroplasty procedures by up to 69%, with an increase in the length of planned hospital stays [16].

Though the Gulf Corporation Countries (GCC) reported almost half of the COVID-19 regional cases, they showcased an excellent model in bringing the outbreak under control in their countries, achieving higher than the global average COVID-19 recovery rates [17]. Countries in the GCC also facilitate substantial financial and material resources for the COVID-19 pandemic, including both preventive and curative resources [17].

By the end of 2020, Dubai, as one of the seven Emirates of the UAE, emerged as a global leader in terms of COVID-19 testing, testing more people per 1000 than any other developed nation and having a lower case fatality percentage, with a case recovery rate of almost 98% [18]. Healthcare in Dubai is provided by a combination of government and private providers. The Dubai Health Authority (DHA) is the government entity that oversees the health sector in the Emirate of Dubai [19]. In addition, there exists the Ministry of Health and Prevention, which is the federal ministry overseeing the UAE healthcare sector as a whole [20]. Both of these entities played a significant role during the pandemic [18]. Regarding healthcare financing, private employers and government play a major role, and the share of out-of-pocket spending remains small, ranging from 10% to 12%. (HASD) [21]. 

Early in the COVID-19 pandemic, it was not clear how healthcare utilization and spending would change. Although one might expect health costs to increase during a pandemic, there were other factors, such as the huge cancellations observed in elective care, driving spending and utilization down [22]. Overall health spending appears to have dropped slightly in 2020, in many parts of the world, for the first time in recorded history [22,23,24].

Similarly, in Dubai, the total current health expenditure in 2020 grew by just 1% compared to a growth rate of 5% in the previous year. However, the health expenditure as a share of GDP increased from 4.7% in 2019 to 5.3% in 2020, the largest increase since 2016 [21,25]. The increase was primarily driven by government spending, including financial assistance to the uninsured individuals and those who had already consumed their allowable insurance limit while being managed for COVID-19. An expansion of public healthcare facilities to cater to increased demand for healthcare services due to COVID-19 was observed. As a result, government spending in 2020 increased to 40% of the total healthcare expenditure in the emirate, compared to 36% in 2019 [21].

While an increasing number of scientific papers have been published on COVID-19 since the start of the pandemic, studies on its impact on service utilization and healthcare spending have not been so plentiful. To our knowledge, no study has comprehensively examined the impact of the COVID-19 pandemic on healthcare services utilization and the spending of services after the pandemic in Dubai, the UAE. The current study tried to estimate the trends of healthcare expenditure and the uptake recorded from disease categories during 2020 and compare the utilization with the previous year (pre-pandemic). It aimed, mainly, to determine the extent of the changes in utilization and spending of healthcare services during the COVID-19 pandemic in Dubai. Accordingly, this study intended to shed light on the consumption pattern of specific healthcare services and the outcomes it drove.

According to our hypothesis, there was no significant difference in healthcare service expenditure and utilization between the pandemic and the pre-pandemic years, as the increase in demand due to the pandemic was counter-balanced by lockdown measures and other government policies.

## 2. Materials and Methods

### 2.1. Data Source and Study Design 

The primary source of data on healthcare utilization came from medical claims data collected by the Dubai Health Authority (DHA) from January 2019 to December 2020. The dataset included 36.87 million and 32.53 million distinct claims for years 2019 and 2020, respectively. Due to the Mandatory Health Insurance system’s requirement, the data captured claim data from all healthcare institutions, including clinics, pharmacies, and hospitals, covering the Dubai population (now estimated at 4.5 million) The data was extracted from the eClaimLink platform of the Dubai Health Authority.

The eClaimLink is the online platform that includes all the claim transaction data for all Dubai-based insurance policyholders, with details of the service provided for each service episode [26]. The data included information at the patient-encounter level regarding diagnosis, investigations, and treatments. The eClaimLink data was queried using SQL for distinct claims by encounter type for all the major diagnostic categories for the years 2019 and 2020. The data on health expenditure by different service categories for all MDCs was extracted and cross-validated with the published Health Accounts of Dubai (HASD) reports for the years 2019 to 2020 [21,25,27,28]. A cross-sectional retrospective study was conducted on the data obtained from eClaimLink. 

### 2.2. Variables 

An outpatient (OP) visit was defined as a visit to a healthcare facility for diagnosis or treatment without an overnight stay in the facility [29]. Hospital admissions were defined as formal admission into a healthcare facility for treatment and/or care that was expected to constitute an overnight stay [29].

Meanwhile, diagnostics were defined as laboratory and imaging procedures related to diagnosis and monitoring [30]. In addition, the current study used the term major diagnostic categories (MDCs) to describe a group of similar diagnosis-related groups, such as all those affecting a given organ system of the body [31].

Only the MDCs with the highest volume (MDC having 5% and above, claim count proportion out of total claims) were considered for the detailed analysis. In the study, ambulatory healthcare was used to describe the services provided to outpatients. Health utilization data for the year 2019 represented the pre-pandemic era, while 2020 data was expressed as the pandemic one. The overall healthcare service was categorized into four groups: outpatient visits; inpatient or admissions; diagnostics (e.g., imaging, pathology, screening investigations); and prescriptions and drugs. 

### 2.3. Outcome Measures 

The primary outcome was the overall change in the spending and utilization of health services, such as visits to the healthcare facility or receipt of diagnostic services between the pre-pandemic and pandemic periods, expressed as a percentage change. The secondary outcome was a change in the services across different disease categories.

### 2.4. Statistical Analysis

The descriptive statistics was used to analyze the health insurance claims data to assess the utilization of healthcare services and compare between pandemic and pre-pandemic year. Using IBM-SPSS for Windows version 25.0 (SPSS Inc., Chicago, IL, USA), data entry, coding, cleaning, weighing, and analysis were completed.

The frequency distribution (using percentages) of total claims by each encounter type was computed across the MDCs. Furthermore, percent change was calculated by dividing the absolute value of the difference between the claim counts of 2019 and 2020 by the counts of 2019 for each service type as shown below:% change of OP visits = (Total OP claim count 2020 − Total OP claim count 2019) × 100/
Total OP claim count 2019

Similarly, healthcare spending along the dimensions of different encounter types was analyzed and percentage change was calculated as shown in the below example:% change of OP expenditure = (Total net paid for OP (2020) − Total net paid for OP (2019)) × 100/
Total net paid 2019

The paired *t*-test was conducted to test the significance differences in the volume of inpatient and outpatient visits and hospital admissions between the pre-pandemic and pandemic years. The statistical significance for the *p* value of ≤0.05 was considered in the study.

## 3. Results

### 3.1. Healthcare Utilization by Major Disease Categories in 2019 and 2020

The medical claim database reported the total claim count including all MDCs to be 36.87 million and 32.53 million for the years 2019 and 2020, respectively. In both years, the highest number of claims was submitted for disease of the respiratory system, followed by disease of the musculoskeletal system and symptoms and signs not classified by disease category. In 2020, the claims for COVID-19 were primarily submitted as codes for special purpose and accounted for 0.2% of the total claim count as shown in Figure 1.

### 3.2. Healthcare Expenditure by Major Disease Categories in 2019 and 2020

Based on medical claims data, the total net amount paid by MDCs was AED 11 bn and 10.3 bn in 2019 and 2020, respectively. In both years, the highest expenditure was on three major categories, namely diseases of the respiratory system, musculoskeletal system, and digestive system. Additionally, in 2020, the codes for special purposes accounted for 2% of the total MDC spending, primarily reflecting the COVID-19 financial burden as illustrated in Figure 2.

### 3.3. Change in Healthcare Utilization (Pandemic versus Pre-Pandemic) 

The overall percent change in utilization by categories of health services revealed that, in 2020, there was a reduction in outpatient visits and inpatient admissions by 27% and 21%, respectively, compared to 2019. Another decrease (13%) was seen in diagnostics (radiology and pathology) and prescription drugs (22%). This data is not shown in a table. 

Table 1 reveals the results of the paired *t*-test for the differences in inpatient and outpatient services between 2019 and 2020. The data indicated that the difference in inpatient admission was high and statistically significant with a *p*-value <0.05. The results of the paired *t*-test indicated that the difference in outpatient visits between 2019 and 2020 was not statistically significant with a *p*-value >0.05. 

Table 2 illustrates the percent change in healthcare utilization for major diagnostic categories along different service lines, namely outpatient, inpatient, diagnostics, and prescriptions. The analysis showed that the outpatient visits and prescriptions decreased across all the MDCs except mental and behavioral disorders, which showed an increase of 8% in outpatient visits and 29% in prescriptions, reflecting the impact of a pandemic on mental health. The admissions to the healthcare facilities were also reduced significantly across various MDCs, ranging from 10% to 44%. Similarly, a downward trend was seen in diagnostics (radiology and pathology) for different MDCs except for neoplasms, which reported an increase of 6%. In 2020, the additional codes were implemented to capture the COVID-19 diagnosis and were reported under Codes for special purposes. Being the new codes, they were excluded from the percentage change calculations. 

### 3.4. Change in Healthcare Spending (Pandemic versus Pre-Pandemic) 

The analysis of the total percentage change in spending across different health service categories revealed that, in 2020, the spending on outpatient visits was lower (22%) compared to 2019; hospital admissions spending also dropped (19%), and, in addition, the expenditure on diagnostic and drugs showed a minor reduction (7% and 4%). 

Table 3 illustrates the percent change in healthcare spending for major diagnostic categories along different service lines, namely outpatient, inpatient, diagnostics, and prescriptions. The analysis showed that during a pandemic, the spending on outpatient and prescriptions for mental and behavioral disorders increased by 11% and 44%, respectively. In addition, an increase in expenditure on diagnostics and drugs for neoplasm was reported despite a corresponding decrease in outpatient and inpatient admissions for the same. For all the other MDCs listed in Table 3, there was an overall decrease in healthcare spending in 2020 compared to 2019 along all the service lines.

## 4. Discussion 

The COVID-19 pandemic has had a profound effect on the performance of healthcare systems throughout the affected countries [32]. The performance of the health system in Dubai was affected by the pandemic, as has been reported in many other settings globally [33,34,35]. Our findings highlight how the pandemic has affected health services utilization and spending.

The current study estimated that all the health service categories witnessed a reduction in utilization between 2019 and 2020, with the greatest decrease encountered in outpatient visits, followed by inpatient admissions (the differences in the volume were statistically significant). This pattern has been reported in many other parts of the world [8,33,34]. Consistent with our findings, early published research by Mahmassani et al. found a 47.2% reduction in general and 66.6% in pediatric emergency visits in 2020 compared to previous months in Lebanon [35]. Our results were in accordance with the findings of the 2020 WHO Pulse survey on service continuity, which showed that around 50% of the surveyed countries reported that some outpatient and inpatient services were markedly interrupted due to government policies. The interruption in many healthcare services could have negatively affected the population health, globally [8]. As per the survey findings, the most frequently disrupted services in the participating countries included routine immunization services (70%), noncommunicable disease (69%), mental health disorders (61%), and cancer diagnosis and treatment (55%). A quarter of countries even showed disruption in life-saving emergency services [8]. 

The current study described how the pandemic has affected health services utilization and spending in Dubai. In the same perspective, the literature has shown a reduction in hospital admissions for all diseases, and, in particular, in cardiovascular and cancer case admissions (which ranged from 35.7% to 10.3% of pre-pandemic time) [36,37,38]. 

The current study findings highlighted that, among the disease categories that witnessed some changes between the pre-pandemic and early in the pandemic, diseases of the respiratory system witnessed the highest drop in all aspects of utilization and spending between 2019 and 2020, though it was expected that there would be an increase in both of them in 2020 compared to 2019 because of COVID-19. This can be explained by the fact that COVID-19 cases were reported separately under codes for special purposes. The clinical presentation of COVID-19 cases was somewhat similar to severe respiratory viral infection, hence accurate coding was a challenge initially [39]. This resulted in the over-diagnosis of COVID-19 cases (reported as codes for special purposes) and the underdiagnosis of respiratory diseases. In addition, protective measures like mask-wearing and sterilization measures also led to a huge drop in other respiratory diseases reported in many parts of the world [40]. 

The disturbance in many services that happened during the early year of the pandemic may have caused the burden of disease to increase, which could influence service utilization patterns in the upcoming years, an outcome that was observed previously with the SARS epidemic in 2003 and that should be taken into consideration in future studies on the impact of COVID-19 [41]. 

Globally, the stress and anxiety caused by the pandemic, along with the high prevalence of mental health problems in many settings, doubled the importance of providing mental health services during such time [42]. Not surprisingly, the present findings showed that mental and behavioral disorders were the only group of diseases that did not show negative change before and after the COVID-19 crisis. An increase in outpatient visits and around one-third of an increase in prescriptions was witnessed from 2019 to 2020. The WHO’s survey indicated that 90% of countries are working to provide mental health and psychosocial support to COVID-19 patients and responders alike [8]. In the same context, the COVID-19 pandemic significantly increased the need for mental health services and spending while simultaneously disrupting the delivery of those services, leaving huge gaps in care for those who needed it in many parts of the world [43]. A range of 10% to 40% of adults reported symptoms of anxiety or depression (with variable impact across communities) [43]. These estimates correspond to results from studies suggesting that persistent psychological distress and anxiety are associated with surviving COVID-19 and with prolonged periods of quarantine, social isolation, and school and work disruptions among the population [44,45,46]. In a similar context, studies reported a global high frequency of workplace violence among healthcare workers [47,48]. Therefore, healthcare systems should focus on protecting the mental health of all healthcare workers. [48].

In terms of healthcare spending, in Dubai, there was a significant expenditure on COVID-19-related services [49]; however, due to the suspension of elective surgeries and disruptions in healthcare delivery, as stated above, there was an overall reduction in healthcare spending across all major diagnostic categories. Further work is needed to quantify and better understand the exact causes and the potential impact of such disruptions. Moreover, though COVID-19 disrupted all health services utilization, outpatient visits were the most affected, followed by medications and total admissions. Nevertheless, the present study revealed that the reduction in overall spending on diagnostics and medication was less than the reduction in utilization spending. This could be possibly explained by the increased usage of high-cost drugs and diagnostics to manage COVID-19 cases. 

The findings of the current study also reported an increase in expenditure on diagnostics and drugs for neoplasm despite a corresponding decrease in outpatient and inpatient admissions for the same. This could be due to high resource consumption during each encounter (along with a reduced frequency of encounters), resulting from increased severity of disease and fear and anxiety because of missed prior consultations. It was noted in the present analysis that the total percent change in drugs was minor compared to the change noted in outpatient visits or inpatient admissions. The number of interactions between patients and providers (both in outpatient and inpatient services) decreased, but those interactions that did occur were resource-intensive, leading to a much smaller drop in the percentage change in medications and diagnostics.

In the UAE, in particular, healthcare costs remained at the center of discussions and challenges that the healthcare sector experienced during the COVID-19 pandemic [50]. Although healthcare costs were expected to increase in response to the pandemic, there were concomitant factors driving the utilization and spending down. This was especially noticeable in major disease categories other than the COVID-19 response and the rising needs of patients within these domains. In a similar context, early in the pandemic, a study suggested that no relationship could be demonstrated between pandemic mortality rates per country and their fiscal policy [51]. In fact, early COVID-19 fiscal policies related most significantly to the country’s creditworthiness, meaning the trend was that countries spent as much as they could [52]. Furthermore, recent data does not demonstrate a link between government spending and improvements in COVID-19 outcomes [3]. 

Research has highlighted that COVID-19 will impact the health services organization and expenditures in the upcoming years, whereas a nationwide study conducted in the United States predicted that economic development would outpace increases in health spending [53]. A McKinsey analysis indicated that the future shape of healthcare services is expected to continue the rapid shift in care from costly acute and post-acute hospitals to the less expensive outpatient facilities brought about by COVID-19. In many regions of the world, this transition has also increased demand for home-based and virtual services [54]. 

### Strengths and Limitations

To our knowledge, this study is the first to estimate the pandemic-related changes in utilization and spending across healthcare services categories and disease categories in Dubai, the UAE. Still, our results are subject to several important limitations. First, the possibility of data collection errors, data accuracy, and reporting bias cannot be ruled out, even though routine data were extracted from the system using standardized procedures. Another limitation was the lack of access to data from other health information systems, which limits data completeness and comprehensiveness in the present analysis. Additionally, the data on emergency services could not be deciphered due to coding issues and was thus excluded from this analysis. Given the varied clinical presentation of the COVID-19 cases, it was a challenge to separate the prescriptions issued only for COVID-19 cases from the database, thus making it difficult to estimate the exact expenditure.

Finally, there might be some underestimation of the utilization of preventive and promotive services, given how the data is reported in the database.

## 5. Conclusions

Our study shows a significant decline in outpatient and inpatient volume in healthcare facilities in Dubai, at all levels, in conjunction with the early pandemic. The findings revealed that diseases of the respiratory system witnessed the highest drop in all aspects of utilization between 2019 and 2020, offset by a significant increase in the cases reported under codes for special purposes. On the other hand, mental and behavioral disorders were the only group of diseases that did not show negative change before and after the COVID-19 crisis. There was a significant expenditure on COVID-19-related services, however, due to the suspension of elective surgeries (that was presented herein as a reduction in admissions) and disruptions in healthcare delivery; the overall impact on healthcare utilization and spending showed a reduction. The current study tried to showcase Dubai’s performance in order to create evidence of those elements of healthcare that may require future attention and action to ensure the uptake of these services by the needed population.

In the event of future pandemics, policymakers and administrators of the health system should benefit from using the lessons and experiences of the COVID-19 pandemic to make better judgments on how to contain and manage the outbreak [55]. Consequently, in the short term, raising undergraduate physicians’ understanding of defensive medicine through proper training will aid in avoiding incurred additional expenses as a result of probable misconduct during times of crisis [56].

In the context of the current findings, we recommend that decision-makers enhance mental health and psychosocial support services as part of strengthening preparedness, response, and resilience to COVID-19 and future public health emergencies. However, many questions remain about the possible causes and impacts of the changes in healthcare utilization documented in the current study, calling for careful analysis and further research with the need to cautiously interpret drivers and impacts of changes.

## Figures and Tables

**Figure 1 healthcare-12-00473-f001:**
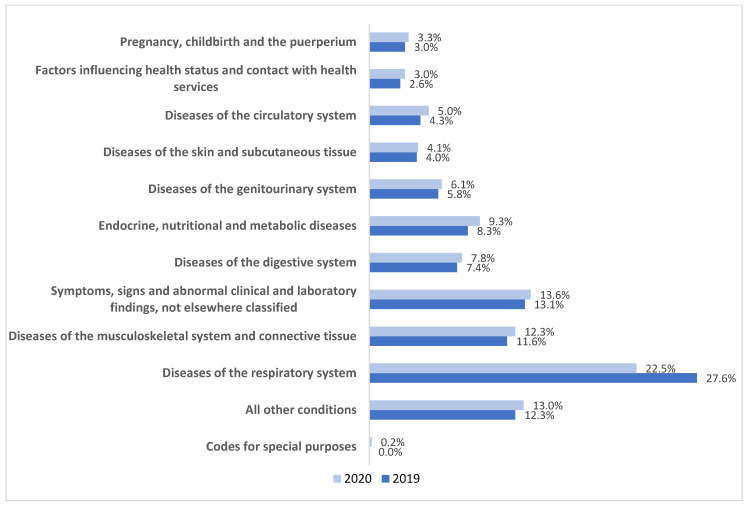
Claim count of major disease categories as a percentage of the total claim count in 2019 and 2020. Source: eClaimLink, DHA.

**Figure 2 healthcare-12-00473-f002:**
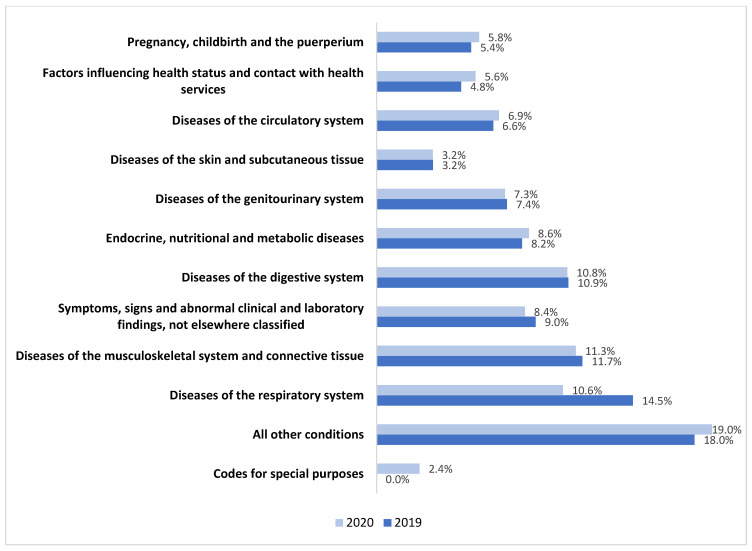
Expenditure on major disease categories as a percentage of the total paid amount in 2019 and 2020. Source: eClaimLink, DHA.

**Table 1 healthcare-12-00473-t001:** Paired *t*-test for the difference in inpatient admissions and outpatient visits between 2019 and 2020.

Service Category	Pre-Pandemic 2019	Pandemic 2020	*t*-Value	* *p*-Value
	Mean (±SD)	Mean (±SD)		
Hospital admissions	269,430 (±353,521)	195,339 (±241,244)	2.660	0.014
Outpatient visits	3,021,246 (±4,592,453)	2,569,995 (±3,494,137)	1.725	0.098

* *p* is significant at <0.05.

**Table 2 healthcare-12-00473-t002:** Percentage change in utilization by the highest volume MDC between 2019 and 2020.

MDC	Outpatient Visits	Hospital Admissions	Diagnostics (Include Pathology)	Drugs/Prescription
Diseases of the circulatory system	−15%	−26%	−5%	−1%
Diseases of the digestive system	−23%	−29%	−9%	−15%
Diseases of the ear and mastoid process	−31%	−40%	−18%	−25%
Diseases of the eye and adnexa	−26%	−42%	−17%	−16%
Diseases of the genitourinary system	−24%	−29%	−11%	−14%
Diseases of the musculoskeletal system and connective tissue	−25%	−24%	−15%	−15%
Diseases of the nervous system	−20%	−25%	−6%	−9%
Diseases of the respiratory system	−36%	−44%	−24%	−36%
Diseases of the skin and subcutaneous tissue	−28%	−39%	−10%	−19%
Endocrine, nutritional, and metabolic diseases	−21%	−21%	−13%	−2%
Mental and behavioral disorders	8%	−22%	−20%	29%
Neoplasms	−9%	−10%	6%	−8%
Pregnancy, childbirth and the puerperium	−16%	−11%	−4%	−14%

MDC Major Diagnostic Categories. Note: COVID-19 cases were reported separately under codes for special purposes MDCs, which were added for reporting purposes in 2020. Since they were new codes, they are not reflected in Table 2.

**Table 3 healthcare-12-00473-t003:** Percentage change in spending by highest-volume MDCs between 2019 and 2020.

MDC	Outpatient Visits	Hospital Admissions	Diagnostics (Include Pathology)	Drugs/Prescription
Diseases of the circulatory system	−13%	−32%	−4%	4%
Diseases of the digestive system	−20%	−29%	−8%	−4%
Diseases of the ear and mastoid process	−31%	−39%	−12%	−25%
Diseases of the eye and adnexa	−20%	−41%	−11%	1%
Diseases of the genitourinary system	−21%	−28%	−9%	−9%
Diseases of the musculoskeletal system and connective tissue	−23%	−23%	−13%	−8%
Diseases of the nervous system	−7%	−37%	−2%	14%
Diseases of the respiratory system	−40%	−37%	−26%	−34%
Diseases of the skin and subcutaneous tissue	−16%	−42%	−5%	7%
Endocrine, nutritional, and metabolic diseases	−16%	−12%	−12%	6%
Mental and behavioral disorders	11%	−16%	−10%	44%
Neoplasms	7%	−15%	27%	30%
Pregnancy, childbirth, and the puerperium	−15%	−12%	−3%	−10%

MDC Major Diagnostic Categories.

## Data Availability

The datasets generated and analyzed during the current study are not publicly available because the data analysis is ongoing, to study variables other than those covered in this study. The data that supports the findings of this study are available upon request, but restrictions apply to the availability of these data. Data is, however, available from the authors upon reasonable request and with permission of the Dubai Health Authority.

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
