# Peer review of "Impact of COVID-19 Pandemic on Utilization of Healthcare Services and Spending Patterns in Dubai, United Arab Emirates: A Cross-Sectional Study"

_healthcare, 2024, doi:10.3390/healthcare12040473_

Round 1
Reviewer 1 Report
Comments and Suggestions for Authors
Authors compiled a manuscript based on their research on topic “Impact of COVID-19 Pandemic on Utilization of Healthcare Services and Spending Patterns in Dubai, United Arab Emir-3 ates: A Cross-Sectional Study”. The authors compiled this manuscript based on the secondary data received from insurance claim data base. It was obvious that during pandemic situation the focus of the entire healthcare system was on the emergency cases of COVID-19 infected patients rather than the patients with chronic diseases. Furthermore, due to lockdown in the entire country, frequent visits to hospitals were restricted to prevent the spread of the COVID-19. There the result shown by the researchers are predictive. Therefore, there is not novelty in the manuscript to be published as a research paper.
Author Response
Feedback reviewer 1:
We thank the editorial team and the reviewers for their careful evaluation of the manuscript, which has helped us to substantially improve our manuscript, titled “Impact of COVID-19 Pandemic on Utilization of Healthcare Services and Spending Patterns in Dubai, United Arab Emirates: A Cross-Sectional Study.” We have provided a point by point response for each suggestion and question raised by reviewers in the text below. We hope our responses are clear and satisfactory.
|
Reviewer’s comment |
Feedback |
Newly added citations |
|
Authors should focus on the impact of pandemic on admissions for cardiovascular diseases. Several paper from other countries were already published on this point and should be cited; |
Response: -As advised, cardiovascular diseases are mentioned in the introduction as one of the most disturbed services due to the pandemic in line 61. -Also the reduction in CVD admissions is mentioned and cited in the discussion in lines 268-271. |
|
|
|
|
|
|
Response: As per the feedback, this limitation has been mentioned in line 343-345. |
|
|
|
As suggested, the impact of pandemic on elective surgery. In particular, oncological surgery is discussed and cited in the discussion in lines 268-271. (in Citations 36, 37). Besides, the magnitude of cancer- related service disturbance including treatment and admission is mentioned in line 262. Reduction in elective surgeries is also highlighted in line 364, 365. |
Di Martino G, Cedrone F, Di Giovanni P, Romano F, Staniscia T. Impact of COVID-19 pandemic on oncological surgery activities: A retrospective study from a southern Italian region. InHealthcare 2022 Nov 21 (Vol. 10, No. 11, p. 2329). MDPI. Søreide K., Hallet J., Matthews J.B., Schnitzbauer A.A., Line P.D., Lai P.B.S., Otero J., Callegaro D., Warner S.G., Baxter N.N., et al. Immediate and long-term impact of the COVID-19 pandemic on delivery of surgical services. Br. J. Surg. 2020;107:1250–1261. doi: 10.1002/bjs.11670. |
|
|
Methods |
|
|
|
Methodology was poor. Authors simply reported a description of data. |
Response: As per the feedback the methodology has been updated and re-written in to Sub-sections |
|
|
Results |
|
|
|
No statistical analysis were performed. It is unclear if authors considered multiple admission and if they extracted only aggregated data. I suggest to test the trend with the appropriate statistical model (joint-point? poisson?). |
Response: Currently, we have major data limitation issues that affected our ability to conduct a more advanced statistical model. Running trend analysis like Poisson is impossible as we only have 2 data points for the years 2019 and 2020. Will consider this advice in future research with trend analysis for data from the upcoming years. However, we ran tested for the significant differences in utilization between the 2 data points using paired t test. |
|
|
|
|
|
|
Reviewer’s comment |
Feedback |
Newly added citations |
|
The abstract is structured appropriately. Minor editing recommendation - The sections should be differentiated in italics for better reading rather than bold.
|
Response: -As suggested the sections were differentiated in italics for better reading rather than bold.
|
|
|
Introduction |
|
|
|
The introduction transposes the research into the topic and formulates the objective of the study at the end. -However, more precise examples on the impact of COVID-19 pandemic on different types of healthcare services in relation to other scientific papers, for e.g. Moldovan, F.; Gligor, A.; Moldovan, L.; Bataga, T. An Investigation for Future Practice of Elective Hip and Knee Arthroplasties during COVID-19 in Romania. Medicina 2023, 59, 314. Please write COVID in caps letters (see line 39) according to the abbreviation used in line 37. Additional comments: Unfortunately, there are no study hypotheses or research questions that should be extracted from the literature analysis and presented in the Introduction section. |
Response: In the introduction:
-Example of the COVID-19 impact on certain services is mentioned as advised in lines 65-68 and is cited accordingly (ref. 16).
- All COVID-19 is written in caps.
- A study hypotheses and research questions (aims) are mentioned in details in lines 111- 113. |
16. Moldovan, F.; Gligor, A.; Moldovan, L.; Bataga, T. An Investigation for Future Practice of Elective Hip and Knee Arthroplasties during COVID-19 in Romania. Medicina 2023, 59, 314. doi: 10.3390/medicina59020314.
|
|
Methods |
|
|
|
In the methodology section, the stages of the research are presented, but at the beginning the methodological stages should be briefly presented, after which they should be treated in subsections.
|
Response: In the methodology section: As advised, all the methods section is enhanced and the methodological stages were presented, along with the subsections as in the following: - Data Source and Study Design - Variables - Outcome Measures - Statistical analysis in lines 150- 175. |
|
|
The three subsections need to be expanded with additional data. Section 2.1 should also indicate more information about participants if available in databases. Section 2.2 should also indicate variables in a more structured way including symbols and the calculation.
methodologies/formulas used. Statistical analysis in paragraph 2.3 should be more detailed presenting data collection (period, source of collection) how was processed, software used, etc.
|
Response: The subsections were enhanced and the requested information added as follows: - information about participants and database is mentioned in lines 117-121
- Formulas are detailed in lines 161- 171. - Statistical analysis section is edited as in lines 154- 172.
- Software used is mentioned in line 156-157 - Further statistical analysis using paired t test is also conducted and stated .(171-174) |
|
|
The fonts used in the figures must be the same as those in the text. The observation is also valid for the tables. Table 1 duplicates the name with the content of line 1, which should be deleted. All heading tables should be on a white background, give up color, and the lines should be black. |
Response: -The fonts used in the figures were unified as those in the text and it will be taken care more by the journal editorial board. -Table titles and headings were edited as advised.
|
|
|
Figures 1-3 are taken from other sources, but original figures would be preferred.
In the methodological section, it would have been desirable to define some variables that would lead to original results and graphic representations, perhaps based on the tabular data presented or some new data.
|
Response: - Original figures are added and all the figures are enhanced, and organized as suggested.
- Variables used in outcome measures are defined in lines 132- 153. - The results section is modified and enhanced accordingly with further analysis as seen in lines 173-214. |
|
|
Regarding the statistical analysis, information on the software used should be provided. |
Response: Information on the software used is mentioned in line 156. |
|
|
Results |
|
|
|
The results are clearly described, but most of them are predictable, indicating a decreasing trend of the aspects studied. Editing recommendation – _ Please move figure 2 before subsection 3.2
|
Response
-Professional editing is done to the whole manuscript.
Editing is performed all over the manuscript.
-Figures are modified as per the feedback and moved accordingly to subsection |
|
|
Discussion |
|
|
|
The discussions interpret the research results and relate them to other results from scientific literature. However, the existence of some hypotheses would have allowed a discussion regarding the degree of validity. |
Response:
-A hypothesis is added in lines 113- 114 and the results validate the rejection of a null hypothesis. It is discussed in details and the discussion section is enhanced as advised. |
|
|
Conclusion |
|
|
|
The conclusions are concise and clear. The references are adequate but need proper editing and can be extended as suggested above, (use same style for names, publications, etc.) and can be extended. |
Response:
-The reference section is subjected to proper editing and unification of the reference style. |
|
|
Reviewer’s comment |
Feedback |
Newly added citations |
|
Introduction |
|
|
|
Lines 40-42: In the worldwide comparison you make, you do not consider European expenditure against the pandemic. It is necessary to address this issue since it was the first epicentre of the pandemic (https://doi.org/10.1016/j.healthpol.2021.11.002).
|
Response: As per the feedback, Europe response is acknowledged in lines 46- 49, and it is cited accordingly.
|
Thomson S, García-Ramírez JA, Akkazieva B, Habicht T, Cylus J, Evetovits T. How resilient is health financing policy in Europe to economic shocks? Evidence from the first year of the COVID-19 pandemic and the 2008 global financial crisis. Health Policy. 2022 Jan 1;126(1):7-15.
|
|
2. No reference is made in the introduction to the risk of increased litigation due to malpractice and the rise of defensive medicine that will have to be addressed by the health systems of all countries worldwide: add some thoughts on this or your paper is incomplete. (https://doi.org/10.1111/1742-6723.13548, https://doi.org/10.1007/s10198-015-0687-8) |
6. Kelly AM. COVID‐19 and medical litigation: More than just the obvious. Emergency Medicine Australasia. 2020 Aug;32(4):703-5.
7.Montanera D. The importance of negative defensive medicine in the effects of malpractice reform. The European Journal of Health Economics. 2016 Apr;17:355-69. |
|
|
Results |
|
|
|
The results need to be rewritten and the graphs and tables modified. First of all, they are too verbose and shown on both figures and text, improve the figures and in the text insert only the relevant results. Tables 1 and 3 are useless, try to compress all results into 2 tables or insert tables 1 and 3 only on the text.
|
Response: The results section is re-written and modified along with the figures and tables. Figure 1 and 2 merged in one graph
Figure 3 and 4 were merged in one graph
Table 1 and 3 findings are mentioned in small write up Additionally paired t test was conducted and Table 1 describes the results |
|
|
Discussion |
|
|
|
In the discussion, the criticality I found in the manuscript emerges. -No mention is made of the future impact on health services of both the reorganization of services and expenditure.
-Lines 260-262: When addressing the impact on mental health, considerations must also be given to the impact of mental health and violence against health workers which has increased due to the pandemic (https://doi.org/10.1016/j.puhe.2023.05.021 and https://doi.org/10.1111/jnu.12794).
-Considerations regarding the risk of increased malpractice due to the reorganization of services must necessarily also be added in order to consider your manuscript more complete.
|
Response:
-The future impact on health services of both the reorganization of services and expenditure is addressed in details in lines 326-332.
-The impact of mental health and violence against health workers is addressed briefly in lines 292-294, and they are cited accordingly (references 47, 48).
-As we are unable obtain data on malpractice to be analyzed in this study, we are unable to discuss this in the discussion section, however, the risk of increased malpractice is addressed in details in the Introduction section in lines 49-53 and was cited accordingly. |
Chernew ME. How to read national health expenditure projections in light of COVID-19: uncertain long-run effects, but challenges for all. Health Affairs Forefront. 2020.
Poisal JA, Sisko AM, Cuckler GA, Smith SD, Keehan SP, Fiore JA, Madison AJ, Rennie KE. National Health Expenditure Projections, 2021–30: Growth To Moderate As COVID-19 Impacts Wane: Health Affairs. 2022 Apr 1;41(4):474-86.
Singhal S, Patel N. The future of US healthcare: What’s next for the industry post-COVID-19 [Internet]. McKinsey & Company. McKinsey & Company; 2022 [cited 2023 Dec 2]. Available from: https://www.mckinsey.com/industries/healthcare/our-insights/the-future-of-us-healthcare-whats-next-for-the-industry-post-covid-19
Rossi MF, Beccia F, Cittadini F, Amantea C, Aulino G, Santoro PE, Borrelli I, Oliva A, Ricciardi W, Moscato U, Gualano MR. Workplace violence against healthcare workers: an umbrella review of systematic reviews and meta-analyses. Public health. 2023 Aug 1;221:50-9. Saragih ID, Tarihoran DE, Rasool A, Saragih IS, Tzeng HM, Lin CJ. Global prevalence of stigmatization and violence against healthcare workers during the COVID‐19 pandemic: a systematic review and meta‐analysis. Journal of Nursing Scholarship. 2022 Nov;54(6):762-71. |
|
Conclusion |
|
|
|
Line 330-332: Using the experiences and lessons learned from the COVID-19 pandemic can help not only health system managers and policy-makers in the possible occurrence of future pandemics, but also, in the immediate term, in increasing the awareness of future doctors regarding these issues and regarding defensive medicine universities in improving the training of 'future doctors' in order to avoid an increase in expenditure at the end of these emergency periods due to possible malpractice following the emergency (https://doi.org/10.1016/j.jflm.2023.102484).
|
Response: As advised, increasing the awareness of future doctors regarding these issues and regarding defensive medicine for undergraduates in universities is addressed and cited accordingly, as seen in lines 358- 360. |
Aulino G, Beccia F, Siodambro C, Rega M, Capece G, Boccia S, Lanzone A, Oliva A. An evaluation of Italian medical students attitudes and knowledge regarding forensic medicine. Journal of Forensic and Legal Medicine. 2023 Feb 1;94:102484. |

Reviewer 2 Report
Comments and Suggestions for Authors
The research aim is to analyze the influence of COVID-19 on healthcare services in terms of utilization and economic status.
The abstract is structured appropriately. Minor editing recommendation - The sections should be differentiated in italics for better reading rather than bold.
The introduction transposes the research into the topic and formulates the objective of the study at the end. However, more precise examples on the impact of COVID-19 pandemic on different types of healthcare services in relation to other scientific papers, for e.g. Moldovan, F.; Gligor, A.; Moldovan, L.; Bataga, T. An Investigation for Future Practice of Elective Hip and Knee Arthroplasties during COVID-19 in Romania. Medicina 2023, 59, 314. doi: 10.3390/medicina59020314.
Please write COVID in caps letters (see line 39) according to the abbreviation used in line 37.
Additional comments: Unfortunately, there are no study hypotheses or research questions that should be extracted from the literature analysis and presented in the Introduction section.
Additional comments:
In the methodology section, the stages of the research are presented, but at the beginning the methodological stages should be briefly presented, after which they should be treated in subsections.
The three subsections need to be expanded with additional data. Section 2.1 should also indicate more information about participants if available in databases. Section 2.2 should also indicate variables in a more structured way including symbols and the calculation methodologies/formulas used. Statistical analysis in paragraph 2.3 should be more detailed presenting data collection (period, source of collection) how was processed, software used, etc.
The fonts used in the figures must be the same as those in the text. The observation is also valid for the tables. Table 1 duplicates the name with the content of line 1, which should be deleted. All heading tables should be on a white background, give up color, and the lines should be black.
Figures 1-3 are taken from other sources, but original figures would be preferred. In the methodological section, it would have been desirable to define some variables that would lead to original results and graphic representations, perhaps based on the tabular data presented or some new data.
Regarding the statistical analysis, information on the software used should be provided.
The results are clearly described, but most of them are predictable, indicating a decreasing trend of the aspects studied. Editing recommendation – please move figure 2 before subsection 3.2
The discussions interpret the research results and relate them to other results from scientific literature. However, the existence of some hypotheses would have allowed a discussion regarding the degree of validity.
Limitations of the study are presented at the end of this section.
The conclusions are concise and clear.
The references are adequate but need proper editing and can be extended as suggested above, (use same style for names, publications, etc.) and can be extended.
The paper needs major revision to be published.
Author Response
Feedback Reviewer 2:
We thank the editorial team and the reviewers for their careful evaluation of the manuscript, which has helped us to substantially improve our manuscript, titled “Impact of COVID-19 Pandemic on Utilization of Healthcare Services and Spending Patterns in Dubai, United Arab Emirates: A Cross-Sectional Study.” We have provided a point-by-point response (attached) for each suggestion and question raised by reviewer 2 in the text below. We hope our responses are clear and satisfactory.
|
Reviewer’s comment |
Feedback |
Newly added citations |
|
The abstract is structured appropriately. Minor editing recommendation - The sections should be differentiated in italics for better reading rather than bold.
|
Response: -As suggested the sections were differentiated in italics for better reading rather than bold.
|
|
|
Introduction |
|
|
|
The introduction transposes the research into the topic and formulates the objective of the study at the end. -However, more precise examples on the impact of COVID-19 pandemic on different types of healthcare services in relation to other scientific papers, for e.g. Moldovan, F.; Gligor, A.; Moldovan, L.; Bataga, T. An Investigation for Future Practice of Elective Hip and Knee Arthroplasties during COVID-19 in Romania. Medicina 2023, 59, 314. Please write COVID in caps letters (see line 39) according to the abbreviation used in line 37. Additional comments: Unfortunately, there are no study hypotheses or research questions that should be extracted from the literature analysis and presented in the Introduction section. |
Response: In the introduction:
-Example of the COVID-19 impact on certain services is mentioned as advised in lines 65-68 and is cited accordingly (ref. 16).
- All COVID-19 is written in caps.
- A study hypotheses and research questions (aims) are mentioned in details in lines 111- 113. |
16. Moldovan, F.; Gligor, A.; Moldovan, L.; Bataga, T. An Investigation for Future Practice of Elective Hip and Knee Arthroplasties during COVID-19 in Romania. Medicina 2023, 59, 314. doi: 10.3390/medicina59020314.
|
|
Methods |
|
|
|
In the methodology section, the stages of the research are presented, but at the beginning the methodological stages should be briefly presented, after which they should be treated in subsections.
|
Response: In the methodology section: As advised, all the methods section is enhanced and the methodological stages were presented, along with the subsections as in the following: - Data Source and Study Design - Variables - Outcome Measures - Statistical analysis in lines 150- 175. |
|
|
The three subsections need to be expanded with additional data. Section 2.1 should also indicate more information about participants if available in databases. Section 2.2 should also indicate variables in a more structured way including symbols and the calculation.
methodologies/formulas used. Statistical analysis in paragraph 2.3 should be more detailed presenting data collection (period, source of collection) how was processed, software used, etc.
|
Response: The subsections were enhanced and the requested information added as follows: - information about participants and database is mentioned in lines 117-121
- Formulas are detailed in lines 161- 171. - Statistical analysis section is edited as in lines 154- 172.
- Software used is mentioned in line 156-157 - Further statistical analysis using paired t test is also conducted and stated .(171-174) |
|
|
The fonts used in the figures must be the same as those in the text. The observation is also valid for the tables. Table 1 duplicates the name with the content of line 1, which should be deleted. All heading tables should be on a white background, give up color, and the lines should be black. |
Response: -The fonts used in the figures were unified as those in the text and it will be taken care more by the journal editorial board. -Table titles and headings were edited as advised.
|
|
|
Figures 1-3 are taken from other sources, but original figures would be preferred.
In the methodological section, it would have been desirable to define some variables that would lead to original results and graphic representations, perhaps based on the tabular data presented or some new data.
|
Response: - Original figures are added and all the figures are enhanced, and organized as suggested.
- Variables used in outcome measures are defined in lines 132- 153. - The results section is modified and enhanced accordingly with further analysis as seen in lines 173-214. |
|
|
Regarding the statistical analysis, information on the software used should be provided. |
Response: Information on the software used is mentioned in line 156. |
|
|
Results |
|
|
|
The results are clearly described, but most of them are predictable, indicating a decreasing trend of the aspects studied. Editing recommendation – _ Please move figure 2 before subsection 3.2
|
Response
-Professional editing is done to the whole manuscript.
Editing is performed all over the manuscript.
-Figures are modified as per the feedback and moved accordingly to subsection |
|
|
Discussion |
|
|
|
The discussions interpret the research results and relate them to other results from scientific literature. However, the existence of some hypotheses would have allowed a discussion regarding the degree of validity. |
Response:
-A hypothesis is added in lines 113- 114 and the results validate the rejection of a null hypothesis. It is discussed in details and the discussion section is enhanced as advised. |
|
|
Conclusion |
|
|
|
The conclusions are concise and clear. The references are adequate but need proper editing and can be extended as suggested above, (use same style for names, publications, etc.) and can be extended. |
Response:
-The reference section is subjected to proper editing and unification of the reference style. |
|

Reviewer 3 Report
Comments and Suggestions for Authors
Dear authors,
This issue is really good to highlight the impact of COVID-19 Pandemic on Utilization of Helthcare Services and Spending Patterns. Congratulations on recognising this and doing something about it.
Your manuscript needs revisions to be made and needs to expand the literature cited in order to also consider this issue from the perspective of the effect these measures will have on health systems in the future.
To be clearer, I will divide my suggestions for improving the manuscript according to sections.
Introduction
1. Lines 40-42: In the worldwide comparison you make, you do not consider European expenditure against the pandemic. It is necessary to address this issue since it was the first epicentre of the pandemic (https://doi.org/10.1016/j.healthpol.2021.11.002).
2. No reference is made in the introduction to the risk of increased litigation due to malpractice and the rise of defensive medicine that will have to be addressed by the health systems of all countries worldwide: add some thoughts on this or your paper is incomplete. (https://doi.org/10.1111/1742-6723.13548, https://doi.org/10.1007/s10198-015-0687-8)
Results
The results need to be rewritten and the graphs and tables modified.
First of all, they are too verbose and shown on both figures and text, improve the figures and in the text insert only the relevant results.
Tables 1 and 3 are useless, try to compress all results into 2 tables or insert tables 1 and 3 only on the text.
Discussion
In the discussion, the criticality I found in the manuscript emerges.
No mention is made of the future impact on health services of both the reorganisation of services and expenditure.
Lines 260-262: When addressing the impact on mental health, considerations must also be given to the impact of mental health and violence against health workers which has increased due to the pandemic (https://doi.org/10.1016/j.puhe.2023.05.021 and
https://doi.org/10.1111/jnu.12794).
Considerations regarding the risk of increased malpractice due to the reorganisation of services must necessarily also be added in order to consider your manuscript more complete.
Conclusions
Linee 330-332: Using the experiences and lessons learned from the COVID-19 pandemic can help not only health system managers and policy-makers in the possible occurrence of future pandemics, but also, in the immediate term, in increasing the awareness of future doctors regarding these issues and regarding defensive medicine universities in improving the training of 'future doctors' in order to avoid an increase in expenditure at the end of these emergency periods due to possible malpractice following the emergency (https://doi.org/10.1016/j.jflm.2023.102484).
Please also consider this issue in your conclusions.
References:
In references 8 and 33 there are also the numbers '12' and '51': please delete them.
Comments on the Quality of English LanguageOnly minor editing of English language is required.
Author Response
Feedback reviewer 3:
We thank the editorial team and the reviewers for their careful evaluation of the manuscript, which has helped us to substantially improve our manuscript, titled “Impact of COVID-19 Pandemic on Utilization of Healthcare Services and Spending Patterns in Dubai, United Arab Emirates: A Cross-Sectional Study.” We have provided a point by point response for each suggestion and question raised by reviewer in the text below. We hope our responses are clear and satisfactory.
|
Reviewer’s comment |
Feedback |
Newly added citations |
|
Introduction |
|
|
|
Lines 40-42: In the worldwide comparison you make, you do not consider European expenditure against the pandemic. It is necessary to address this issue since it was the first epicentre of the pandemic (https://doi.org/10.1016/j.healthpol.2021.11.002).
|
Response: As per the feedback, Europe response is acknowledged in lines 46- 49, and it is cited accordingly.
|
Thomson S, García-Ramírez JA, Akkazieva B, Habicht T, Cylus J, Evetovits T. How resilient is health financing policy in Europe to economic shocks? Evidence from the first year of the COVID-19 pandemic and the 2008 global financial crisis. Health Policy. 2022 Jan 1;126(1):7-15.
|
|
2. No reference is made in the introduction to the risk of increased litigation due to malpractice and the rise of defensive medicine that will have to be addressed by the health systems of all countries worldwide: add some thoughts on this or your paper is incomplete. (https://doi.org/10.1111/1742-6723.13548, https://doi.org/10.1007/s10198-015-0687-8) |
6. Kelly AM. COVID‐19 and medical litigation: More than just the obvious. Emergency Medicine Australasia. 2020 Aug;32(4):703-5.
7.Montanera D. The importance of negative defensive medicine in the effects of malpractice reform. The European Journal of Health Economics. 2016 Apr;17:355-69. |
|
|
Results |
|
|
|
The results need to be rewritten and the graphs and tables modified. First of all, they are too verbose and shown on both figures and text, improve the figures and in the text insert only the relevant results. Tables 1 and 3 are useless, try to compress all results into 2 tables or insert tables 1 and 3 only on the text.
|
Response: The results section is re-written and modified along with the figures and tables. Figure 1 and 2 merged in one graph
Figure 3 and 4 were merged in one graph
Table 1 and 3 findings are mentioned in small write up Additionally paired t test was conducted and Table 1 describes the results |
|
|
Discussion |
|
|
|
In the discussion, the criticality I found in the manuscript emerges. -No mention is made of the future impact on health services of both the reorganization of services and expenditure.
-Lines 260-262: When addressing the impact on mental health, considerations must also be given to the impact of mental health and violence against health workers which has increased due to the pandemic (https://doi.org/10.1016/j.puhe.2023.05.021 and https://doi.org/10.1111/jnu.12794).
-Considerations regarding the risk of increased malpractice due to the reorganization of services must necessarily also be added in order to consider your manuscript more complete.
|
Response:
-The future impact on health services of both the reorganization of services and expenditure is addressed in details in lines 326-332.
-The impact of mental health and violence against health workers is addressed briefly in lines 292-294, and they are cited accordingly (references 47, 48).
-As we are unable obtain data on malpractice to be analyzed in this study, we are unable to discuss this in the discussion section, however, the risk of increased malpractice is addressed in details in the Introduction section in lines 49-53 and was cited accordingly. |
Chernew ME. How to read national health expenditure projections in light of COVID-19: uncertain long-run effects, but challenges for all. Health Affairs Forefront. 2020.
Poisal JA, Sisko AM, Cuckler GA, Smith SD, Keehan SP, Fiore JA, Madison AJ, Rennie KE. National Health Expenditure Projections, 2021–30: Growth To Moderate As COVID-19 Impacts Wane: Health Affairs. 2022 Apr 1;41(4):474-86.
Singhal S, Patel N. The future of US healthcare: What’s next for the industry post-COVID-19 [Internet]. McKinsey & Company. McKinsey & Company; 2022 [cited 2023 Dec 2]. Available from: https://www.mckinsey.com/industries/healthcare/our-insights/the-future-of-us-healthcare-whats-next-for-the-industry-post-covid-19
Rossi MF, Beccia F, Cittadini F, Amantea C, Aulino G, Santoro PE, Borrelli I, Oliva A, Ricciardi W, Moscato U, Gualano MR. Workplace violence against healthcare workers: an umbrella review of systematic reviews and meta-analyses. Public health. 2023 Aug 1;221:50-9. Saragih ID, Tarihoran DE, Rasool A, Saragih IS, Tzeng HM, Lin CJ. Global prevalence of stigmatization and violence against healthcare workers during the COVID‐19 pandemic: a systematic review and meta‐analysis. Journal of Nursing Scholarship. 2022 Nov;54(6):762-71. |
|
Conclusion |
|
|
|
Line 330-332: Using the experiences and lessons learned from the COVID-19 pandemic can help not only health system managers and policy-makers in the possible occurrence of future pandemics, but also, in the immediate term, in increasing the awareness of future doctors regarding these issues and regarding defensive medicine universities in improving the training of 'future doctors' in order to avoid an increase in expenditure at the end of these emergency periods due to possible malpractice following the emergency (https://doi.org/10.1016/j.jflm.2023.102484).
|
Response: As advised, increasing the awareness of future doctors regarding these issues and regarding defensive medicine for undergraduates in universities is addressed and cited accordingly, as seen in lines 358- 360. |
Aulino G, Beccia F, Siodambro C, Rega M, Capece G, Boccia S, Lanzone A, Oliva A. An evaluation of Italian medical students attitudes and knowledge regarding forensic medicine. Journal of Forensic and Legal Medicine. 2023 Feb 1;94:102484. |

Round 2
Reviewer 1 Report
Comments and Suggestions for Authors
NA
Reviewer 2 Report
Comments and Suggestions for Authors
The authors responded to all the reported observations.
Reviewer 3 Report
Comments and Suggestions for Authors
I thank the authors for responding to all my feedback. For me it is now suitable for publication.